# Problems and Technical Issues in the Diagnosis, Conservation, and Rehabilitation of Structures of Historical Wooden Buildings with a Focus on Wooden Historic Buildings in Poland

## Wojciech Terlikowski

Institute of Building Engineering, Faculty of Civil Engineering, Warsaw University of Technology,
00-637 Warsaw, Poland; wojciech.terlikowski@pw.edu.pl

**Abstract:** This article discusses the problems and technical issues related to the conservation and rehabilitation of historic wooden structures. Showing the history of wooden construction in Europe and Poland, it analyzes the technical solutions used for the timber structure of buildings. The author highlights the leading technical and conservation issues arising from the principles of conservation of wooden heritage buildings, taking into account the role, importance, and proper implementation of diagnostics preceding conservation engineering activities. The article discusses interdisciplinary diagnostics developed by the author, an original method for assessing the revitalization capability of a historical building, which analyzes non-technical issues, including those relevant to conservation, in addition to the technical aspects of assessing the technical condition of a building. Based on engineering practice, problems and technical issues that often occur in the conservation process of wooden buildings, as well as in the broader process of their revitalization, are presented. They concern various engineering activities that must be implemented in this process according to the principles discussed.

**Keywords:** assessment of the revitalization capability of a historic building; wooden structures; historical structures

## 1. Introduction

### 1.1. Specifics of Diagnostic and Preservation Activities for Wooden Historic Buildings

Historic wooden buildings are a very important part of the world's cultural heritage. They are an integral part of the cultural landscape and have been integrally connected with it since the beginning of human history. Diagnosis of the technical condition of wooden historic buildings and related conservation activities, including the entire process of rehabilitation of the main structural elements, are very important issues, even crucial, due to the unique nature and specificity of these buildings. Proper diagnosis of the technical condition of a historic building is fundamental to all rehabilitation and conservation activities. The uniqueness of the approach to wooden historic buildings is related to many aspects specific to wooden construction, including:

- The unique historical and cultural heritage of wooden architecture and construction, which very often testifies to the cultural identity of regions, resulting in the originality of forms, shapes, and technical solutions. This often constitutes the uniqueness and originality of the historic building and is associated with local construction, art, and craftsmanship, which often use original and individual technical solutions that today are forgotten or even unknown [1].
- The diversity of the heritage of wooden buildings, including the variety of forms, layouts, and architectural and structural systems and solutions [2].
- The physical and mechanical properties of the wood used in buildings and the associated woodworking technologies, as well as the technical and structural solutions resulting from these properties [3].

- The susceptibility of wood to destructive influences—environmental, biological, fires, improper use, lack of maintenance, and warfare.
- Uneven degradation of structural wood elements resulting from local sources and varied causes of deterioration [4].
- Difficulties in determining an unambiguous assessment of the technical condition of the building due to the impossibility of conducting a full survey of the main structural elements without destroying the historic sheathing and architectural elements.

Investigations and technical condition diagnosis of wooden structures of historic buildings, as well as the related rehabilitation measures, are more complex and complicated than those of other existing buildings. Taking into account the essence of the monument, it is necessary to determine in detail the valorizing features of the rehabilitated building in terms of architectural form and details, structural and construction systems, construction technology, and original materials, as well as the associated artistic, historical, and cultural value. In addition to technical issues, it is therefore also necessary to analyze non-technical aspects affecting the value of the rehabilitated building, which is directly related to the principles of sustainable development. Rehabilitation and conservation of wooden historic buildings follow directly from the definition and essence of the principles of sustainable development. This article presents the main problems and technical, as well as non-technical, issues related to the diagnosis of the technical condition of wooden historic buildings and conservation and rehabilitation activities in the context of the conservation guidelines developed by the International Council on Monuments and Sites (ICOMOS). The purpose of the article is to demonstrate the complexity of technical and non-technical problems in diagnosing the technical condition of wooden historic buildings and the resulting engineering activities. Additionally, an original methodology developed by the author for multi-criteria evaluation of the revitalization capacity of historic buildings is described, a method that can be used in the diagnosis of these buildings and which fulfils a number of technical and non-technical criteria, including criteria relevant to wooden buildings, while taking into account the principles of sustainable development.

*1.2. Historical Outline Proving the Historical and Cultural Value of Wooden Architecture and Construction*

Wooden construction is one of the oldest forms of building used worldwide. This type of construction developed in areas of permanent settlement located in or near forested areas due to the easy access to building material, the possibility of reproducing it, and the ease of processing resulting from the physical characteristics of wood. In the initial phase, wooden shacks were probably built, which in later periods were transformed into the first forms of frame structures. The earliest traces of wooden construction date back to the Neolithic period (about 9000–3400 BC in a worldwide context and about 5200–2300 BC in Poland). These were mainly buildings with walls made of wooden braids made of branches that were covered with clay and stiffened by wooden posts driven directly into the ground. Thousands of archaeological sites have been discovered in Europe testifying to this building construction [5,6]. The existence of such buildings is confirmed in the work of Vitruvius [7], who described primitive wooden structures that did not require specialized tools to build and were thus widely used. Also shared in these periods were dugouts, strengthened with wooden roof structures supported by a single post [8]. Based on these structures, frame structures were developed at the end of the Neolithic, which were further developed during the Bronze Age (c. 3400–1200 BC worldwide and c. 2300–700 BC in Poland) and early Iron Age (c. 1200 BC to 0 worldwide and c. 700 BC to c. 1st century AD in Poland). Together with timber frame structures, the Iron Age saw the spread of techniques for building wooden houses from horizontally stacked wooden logs. This includes log structures, post-plank structures, inter-pile structures, and those made of logs set vertically and driven into the ground that are referred to as palisade structures, the development of which took place in the Middle Ages [9–14].

Some of Poland's first materially documented wooden buildings, forming a compact urban layout, date to the early Iron Age. This is evidenced by studies of wooden logs preserved in Biskupin that show that 50 percent of the oak trees used in the construction of the defensive fortress there were felled in 748 BC, when the construction of the fortress buildings in Biskupin began [12]. The study also showed that the houses of the older phase of the fortress were made using a post and plank construction method with a roof post and a ridge beam, which was covered with a gabled, thatched roof. The roof structure consisted of rafters supported by a ridge beam and cap beams in the longitudinal walls. The wall structures consisted of rectangular posts and planks (Figure 1). Thus, it was a classic structure, still used today in wooden construction [12,13].

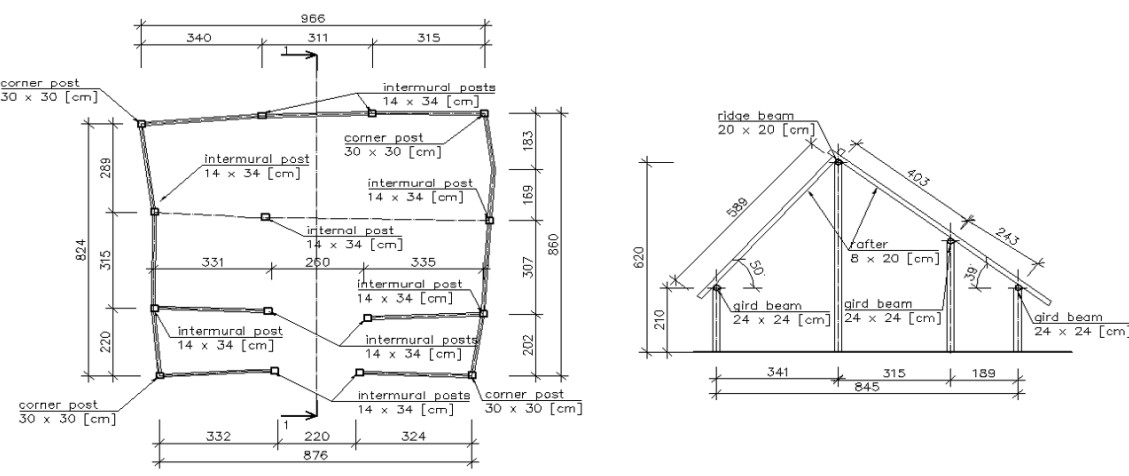

**Figure 1.** A projection and cross-section of an actual building in Biskupin [12,13].

Archaeological research conducted on Wolin Island [14,15] and in its vicinity has confirmed that in the early Middle Ages there were already timber-framed buildings in this area, realized in the various construction techniques mentioned earlier. The earliest fully preserved monuments of wooden construction in Europe date from the Middle Ages. These are usually churches. The oldest fully preserved monument is a palisaded church in the English town of Greenstead in the county of Essex, dating from the 9th to the 11th century [16]. In Scandinavia, in Western Europe (England, France, and Germany), and also in Eastern Europe [16] (such as in Karpacz in Poland), churches [17], multi-storied townhouses, residential buildings, and rural buildings of frame and post-plank construction were built at that time. In the more recent period, wooden structures in central and western Europe were enriched with new ornamental details, but their development was reduced due to fire restrictions in larger estates.

A characteristic feature of wooden construction in Poland (from the turn of the 13th and 14th centuries to the mid-17th century) was, on the one hand, the typification of planning, spatial, and construction solutions resulting from the standardization of building forms and solutions, while on the other hand, architectural and construction solutions slowly began to acquire individual characteristics for particular regions. This was due to local building and craft traditions and the regionalization of carpentry techniques. It can be assumed that the features inherent to wood received attention in the 17th century (probably already in the first half of the century), with unique forms of details and spatial solutions that were unattainable with other materials then explored. The originality of these solutions is evidenced by the fact that they are mainly absent outside of Poland [18]. Such forms were represented by nobleman's mansions, churches, and synagogues. It was also at that time (if not much earlier) that wooden jetties formed, unique in other construction and enriched with knee braces and hangers; this period also saw city houses, town halls, granaries, churches, synagogues, Orthodox churches and rural cottages (18th–19th centuries) begin to develop massively [18–20]. Buildings were erected in both frame

structures (mostly buildings with relatively few vertical structural elements) and log structures. These later were more common and used for buildings of a more monumental nature. In the 18th and 19th centuries, timber frame structures with a small number of columns began to disappear. They were replaced in western and northern territories by timber frame structures with numerous and regularly spaced columns and diagonal bracings. In other areas, log structures with simplified joints prevailed. Currently, such structures are found mainly in southern and eastern Poland, extending to the western border of the Vistula basin [9,10].

The uninterrupted development of timber construction continued until the mid-19th century in the Alpine and Carpathian countries, particularly in northeastern Europe and northern Asia. It was mainly timber construction used for residential purposes (e.g., the palace in Kolomenskoye in the 17th century), defense (the defense complex in Tobolsk in the 17th century), and religious purposes (especially Orthodox churches—characterized by an extraordinary richness of forms and spatial arrangements). Since the mid-19th century, new solutions for wooden construction have been sought primarily in North America, with plank-frame structures as a starting point [16]. In the second half of the nineteenth century, European countries were characterized by a need to revive native architectural traditions. The search for a national style turned to wooden construction. The opinion at that time that those structures had a national character found expression in the architecture of the late nineteenth and early twentieth centuries. In Poland, among others, this can be seen in the Zakopane style and, later, the mansion style [18].

Archaeological studies carried out in various parts of the world (including Biskupin and Wolin) show that the basic systems and structural arrangements used in wooden buildings for nearly three thousand years have not changed significantly and are still in use today [12–15]. This may give the impression that wooden structures are very well understood in terms of the material properties of the wood used for construction, architectural–functional systems, load-bearing structures, structural elements, and their static and dynamic response. This is true but does not exclude possible technical problems related to maintenance, repair, or overall reconstruction. The primary load-bearing system currently used in wooden buildings in urban development is the post and beam system, a type of framing system. This system most often appeared as the so-called half-timbered system—a type of timber frame wall filled with clay mixed and worked with chaff, sawdust, or shavings or cast on a braid of osier vittles or reed stalks. Another type of infill of half-timbered walls was brick masonry. Such systems have been used from the Middle Ages to the present day throughout Europe and even in Middle Eastern countries [21–23]. There are also log structures in urban buildings, known since ancient times, and the aforementioned post-plank structures.

A severe problem with construction and wooden architecture in Poland is the lack of continuity in the historical aspect. This has been influenced, among other things, by the periods where Poland was divided as a result of partitions, the damage done during wars (especially during the Polish–Swedish War (1655–1660) and World War II), territorial changes after World War II, as well as improper use, lack of proper protection, and maintenance. This has directly impacted the fact that preserved monuments of wooden construction and architecture, which are an essential part of national cultural heritage, are now rare on a national scale. Nowadays, conservation efforts to preserve historic wooden buildings are more commonly part of a more complex process of revitalizing an area in crisis. The central point of this area is usually a historical building. To properly carry out the conservation of historic wooden buildings, it should be preceded by a detailed interdisciplinary diagnosis [24,25]. This should result in the determination of the technical condition of the entire building, as well as the individual structural elements that make up the load-bearing system, the determination of the physical and mechanical properties of the materials from which the structural and architectural elements are made, and a detailed architectural and construction survey that includes significant architectural and decorative details with indications of existing damage. Detailed survey, descriptive, and graphic

documentation (photographic, 2D, 3D laser scanning) of historic wooden buildings, which precedes conservation activities, is of great importance due to their unique character in terms of architectural form, decorative details, architectural–functional systems, and strictly structural solutions relating to the entire structural system and the individual elements that make it up.

*1.3. Principles of Conservation, Research, and Engineering Activities on Historic Wooden Buildings with a Discussion of Their Specifics*

In December 2017, the "Principles for The Conservation of Wooden Built Heritage" [26] document was developed and adopted by the IIWC in Delhi. These principles were created to update the "Principles for the Preservation of Historic Wooden Structures" document adopted by ICOMOS at the 12th General Assembly in Mexico in October 1999. Their update was initiated in Guadalajara, Mexico (2012) and Himeji, Japan (2013), and then continued in Falun, Sweden (2016). The document takes into account the general principles of the Venice Charter (1964), the Amsterdam Declaration (1975), the Burr Charter (1979), the Narra Document on Authenticity (1994), and the ICOMOS and UNESCO doctrines on the protection and conservation of wooden built heritage [27]. This document aims to formulate basic principles and practices applicable internationally to the broadest possible extent for the protection and conservation of wooden built heritage and its cultural significance. It regulates strict conservation and engineering activities—protection, strengthening, and repair. To preserve and protect historical wooden structures, the following principles have been formulated [26]:

- Recognition of the importance of wooden structures from all periods as part of the cultural heritage of the world;
- Consideration of the great diversity of historic wooden structures;
- Consideration of the different construction techniques, species, and quality of wood used to complete them over the centuries;
- Recognizing that wooden building heritage is a testament to the skill of craftsmen and builders and their knowledge, which results from tradition, culture, and generational transmission;
- Realizing the constant evolution of cultural values over time and the need to periodically review how they are defined and how authenticity is determined, taking into account changes in attitudes and perceptions of these values;
- Respect for different local traditions, construction practices, and conservation approaches, taking into account the diversity of methodologies and techniques that may apply to protection;
- Recognizing that timber structures contain a valuable record of chronological data in the overall building or structure;
- Taking into account the excellent preservation of wooden structures in terms of their seismic resistance;
- Recognizing that structures, partially or entirely made of wood, are vulnerable to deterioration and degradation of the material under changing environmental and climatic conditions caused by fluctuations in temperature and humidity, exposure to solar radiation, fungal and insect infestation (biological aggression), wear and cracking, fire and other exceptional impacts and natural disasters, and destructive human activities;
- Recognizing the need to take into account the disappearance of historical wooden structures due to their exceptional vulnerability to destruction, improperly applied craftsmanship or a lack of capacity and knowledge of traditional structural design and technology, improper use, and maintenance;
- Consideration of the incredible variety of activities undertaken and treatments necessary for the preservation and conservation of these heritage resources;

- Recognizing the importance of community participation in preserving wooden building heritage, its relationship to social and environmental transformation, and its role in sustainable development.

Some of these activities go beyond the commonly used methods of engineering work on historic buildings made with other technologies (e.g., traditional technology such as masonry) [28]. This can pose additional problems for contractors and engineers involved in preserving or revitalizing a historic building. To clarify the scope of the activities that need to be performed in the correct process of conservation of historic wooden buildings, while taking into account their unique character, specific actions have been postulated [26], which are described in the following subsections.

### 1.3.1. Inspection, Survey, and Documentation

Inspection of the technical condition of a wooden building is a necessary activity. It should include a complete diagnosis of a historic wooden building, a thorough analysis of the technical condition of the structure and the materials used to construct the object, and an analysis of their parameters and properties. This also means thorough recognition of the construction techniques used to erect the object and technical solutions that may no longer be used today. This technical diagnosis of wooden structures should be carried out before any building interference or engineering and construction work is started. In cases of apparent necessity, rescue measures should be taken. Minor interventions resulting from ongoing maintenance are also possible. Such inspection may not be sufficient to properly assess the structure when other materials obscure it in the building. When the value of these materials permits, it may be recommended that they be temporarily and locally dismantled to allow examination, but only after the existing elements have been thoroughly documented. It should be noted that all explorations should be carefully documented. When restoring the details and layers removed during the uncovering, special care should be taken to restore the original solutions, both in terms of materials and technology, so as not to give the structure new, inauthentic features.

All research and conservation activities should be carefully and thoroughly documented. A further separate task is a detailed survey of the building, including architectural details that should be separately surveyed and registered. It should include the damage and pathologies that occur, which should additionally be documented in the form of annotations on the survey that are taken with photographs or drawings and described in detail. It should be noted that in light of the abovementioned recommendations, the survey of a historic building, especially a wooden one, should be carried out in an accurate, detailed manner that takes into account the technical principles of making such documentation, as well as its historical, cultural, and aesthetic value. Considering this, it is good to correlate the survey with photographic documentation to reflect the diagnosed building fully. Currently, the most complementary form of realization of such a survey is 3D laser scanning, which allows the documentation to be digitized and creates a 3D model of the surveyed object [29], thus providing a record of images and structures for all scanned surfaces. Laser scanning can be carried out repeatedly, making it possible to document successive stages of engineering and conservation activities while additionally providing verification of the correctness and accuracy of the work carried out. An additional advantage of scanning is the possibility of examining the digitized object anywhere under laboratory conditions (e.g., in a computer lab or at home). Dimensional accuracy determines the size of all the damage present and its exact location. This is especially important in repeated diagnostic examinations when comparing objects of interest, damage, and locations. The undoubted value of documentation resulting from 3D laser scanning is the creation of a base for the construction of a numerical model of a historic building that accurately reflects its form, its architectural and structural layout, architectural details and ornamentation, and the details of window and door carpentry (along with details, cross-sections), thus providing the possibility of making cross-sections and projections in any place and at any level. A peculiarity of wooden buildings is that, most often, the structural system of the building is

hidden under the surface layers seen on the plane of the facade, ceilings, or interior walls. Performing diagnostic examination of structural elements is often impossible, and it is also difficult to select representative sites for them.

An example implementation of the above is the use of 3D laser scanning in diagnostic work conducted at the palace in Pawlowice, Poland, in which laser scanning, in addition to creating a detailed inventory and surveyed object (its digitization), also contributed to determining the magnitude of the maximum deflections of the ceiling in one of the rooms at risk, thus indicating the locations of test holes through which material samples could be obtained for subsequent tests (enabling calculations and numerical analysis of moisture and mycological conditions) [30].

In some cases, it is possible to identify the structure hidden inside the walls and ceilings using thermal imaging cameras. Exemplary case studies of properly developed inspection, survey, and documentation projects are presented in [31,32].

### 1.3.2. Analysis and Evaluation

According to the ICOMOS document [26], the main goal of conservation is to maintain the authenticity of the historic building, that is, its architectural–functional layout and the resulting structural system (load-bearing system), the materials used, technical solutions (including connections), preservation of the integrity of the entire building, historic architectural, and cultural values, taking into account the changes that have occurred in the history of the building. To this end, it is necessary to preserve, as much as possible, all the features that define the character of a monument, which are specified in the cited document [26] as:

- The overall structural system;
- Non-structural elements such as facades, divisions, and stairs;
- Surface appearance and features;
- Decorative carpentry treatment;
- Traditional solutions and techniques;
- Structural materials, including their quality and unique features.

Examples that meet the above recommendations are usually sites included on the UNESCO World Heritage List, which in in Poland includes the Church of Peace, built in the 2nd half of the 17th century (in Jawor, Lower Silesia province), and the Gothic Church of the Assumption of the Blessed Virgin Mary, built at the end of the 14th century (in Hoczów, Lesser Poland province) [33].

### 1.3.3. Interventions

All interventions in the existing structure of historic wooden buildings should consider that the first aim of preservation and conservation is to preserve cultural heritage's historical authenticity and integrity. The goal to be pursued is minimal intervention in the fabric of the historic wooden structure. If there is intervention, the historic structure should be treated as a whole. Before proceeding with any conservation and engineering activities, including securing, reinforcing, or repairing, a general conservation strategy for the whole historic structure should be developed that considers its cultural value. Such a strategy is usually multidisciplinary and even interdisciplinary, considering the participation of experts from many scientific disciplines [30]. All engineering activities should therefore meet the following parameters [26]:

- The activity is applied to the minimum extent necessary;
- The activity is based on traditional construction and craft practices;
- The activity is reversible;
- The activity does not prejudge or impede the nature of future restoration work where possible;
- The activity does not impede future access to the information contained in the structure, both visible and hidden.

All repairs to wooden structural elements should be carried out traditionally using carpentry techniques (if this is for any reason impossible, it is permissible to use modern solutions that work analogously) and traditional materials, or those compatible with them. The principle of compatibility implies using wood analogous to the original, or with the same physical and mechanical properties, for repair or reconstruction. Any technical and material solution should consider environmental conditions—environmental impacts that represent a potential for destruction (snow, rain, solar radiation, wind, biological aggression) and social conditions related to local social and livelihood issues in the nearby community.

In exceptional cases, when the need for repair involves major structural elements or other architectural or structural elements of exceptional historical, cultural, or aesthetic significance, thus making it impossible to repair them in other traditional ways, it is possible to use contemporary materials and technologies, which should be clearly marked and distinguished from the original materials and solutions. Examples of such materials include modern composites based on synthetic resins that are reinforced with steel rods, polyester or fibers, and carbon strips [34–36].

The effect of all engineering activities on a historic wooden building should be to leave as many original wooden elements as possible. The replacement must respect the above rules when there is a need to replace a component or its part. The wood to be repaired should, as far as possible, possess the following properties [26]:

- Is the same species as the original;
- Has a moisture content consistent with the original wood;
- Has grain characteristics similar to that of the wood to be replaced in places where it will be visible;
- Can be worked with similar craft methods and tools as the original.

New elements or new parts of elements should be discreetly labeled to distinguish them later. According to ICOMOS recommendations, a survey of all materials used during repairs and treatments should be made (stated in the Venice Charter and the ICOMOS document "Principles for the Recording of Monuments, Groups of Buildings, and Sites") [27]. All relevant documentation, including characteristic samples taken from materials no longer useful for further use or elements removed from the structure, and essential information about the entire building should be included. A variety of charts developed by various research and conservation centers describing the technical condition of historic wooden buildings can be helpful here [37–40].

### 1.3.4. Monitoring and Maintenance

Proper multifaceted monitoring, appropriate maintenance, and being used for the intended purpose are critical factors in the preservation of historic wooden building structures and their historical significance. According to an ICOMOS document [26], a consistent strategy should be established for the regular monitoring and ongoing maintenance of wooden building structures to delay the need for major interventions and to ensure the continued protection of wooden heritage and its cultural significance. Monitoring should be carried out during and after each intervention to determine the effectiveness of the methods used and to ensure the wood's long-term performance, as well as that of all other materials used. Records of all work should be collected, and monitoring should be kept as part of the historical documentation of the structure.

### 1.3.5. Preserving Historic Forest Reserves and Conducting Education and Training

According to the principles of sustainable development, efforts should be made to establish and protect forest reserves from which suitable timber can be obtained for preserving and repairing historic wooden structures. It is also recommended that educational activities and training programs be organized and further developed on the subject in question—preservation, protection, and conservation of historical wooden structures that takes into account the principles of sustainable development.

*1.4. Problems, Technical Issues, and Their Examples Occurring in the Diagnosis, Conservation, and Rehabilitation of Wooden Structures of Historic Buildings in Poland*

1.4.1. Difficulties in Diagnosis

The diagnosis of wooden structures of historic buildings is a challenging and complex process due to the physical properties of wood and the various associated forms of possible degradation, such as those associated with moisture (or changes in moisture content) and its effect on wooden elements and the possibility of destructive effects from the environment (rain, snow, frost, wind, solar radiation, biological corrosion caused by fungi and insects, varying temperature conditions, fire, etc.). Issues related to the diagnostic process have been discussed in previous chapters. It is common for structural elements to be hidden under the facade and finish layers in historic wooden buildings. For this reason, without a comprehensive uncovering of the structural elements, assessing its technical condition based on the performance of local excavations may often not be reliable. Degradation does not develop uniformly and holistically in the building and is very often local. Thus, different forms of decay and damage to existing structural elements can occur in the building in different places. Excavations during diagnostic activities are usually local and may not provide a clear and accurate picture of the entire structural system. An example of this can be found in the case of a wooden manor house in the suburbs of Warsaw. The structure was classified as being in medium condition after diagnostics but was found to actually have a varied structure in terms of technical condition during renovation activities, with large parts of the structure in terrible condition (Figures 2–4).

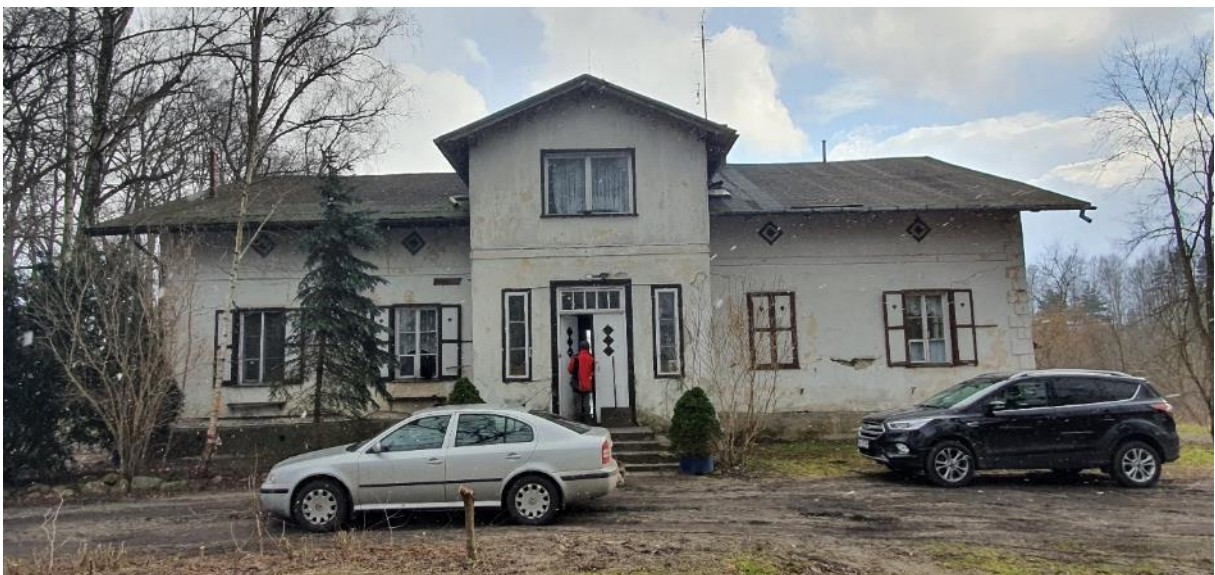

**Figure 2.** The wooden manor house in the suburbs of Warsaw.

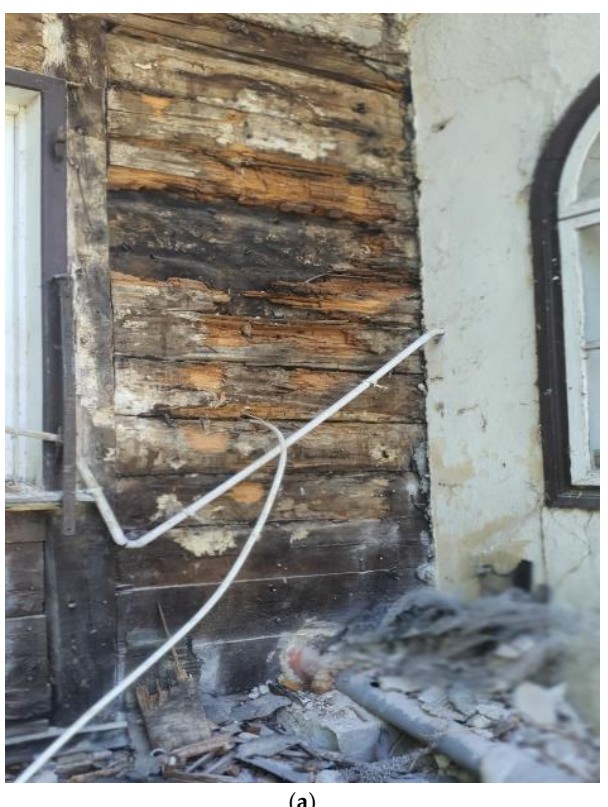
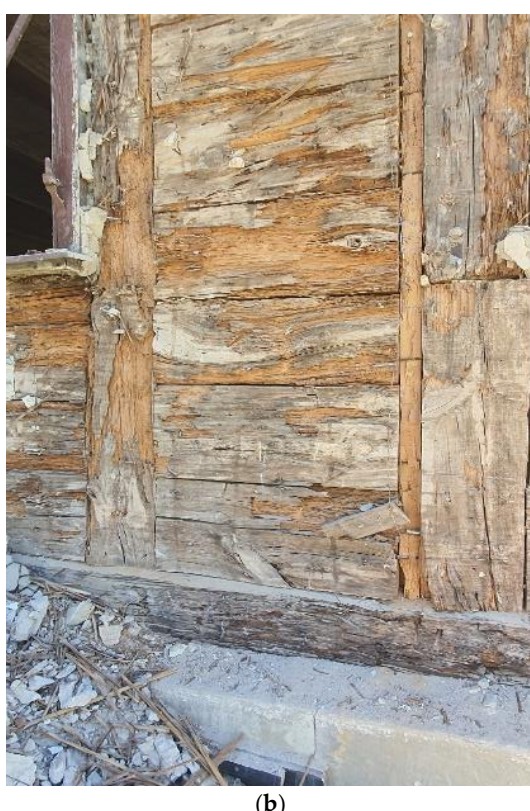

(**a**)                                                    (**b**)

**Figure 3.** (**a**) Corroded front wall; (**b**) corrosion caused by insect aggression.

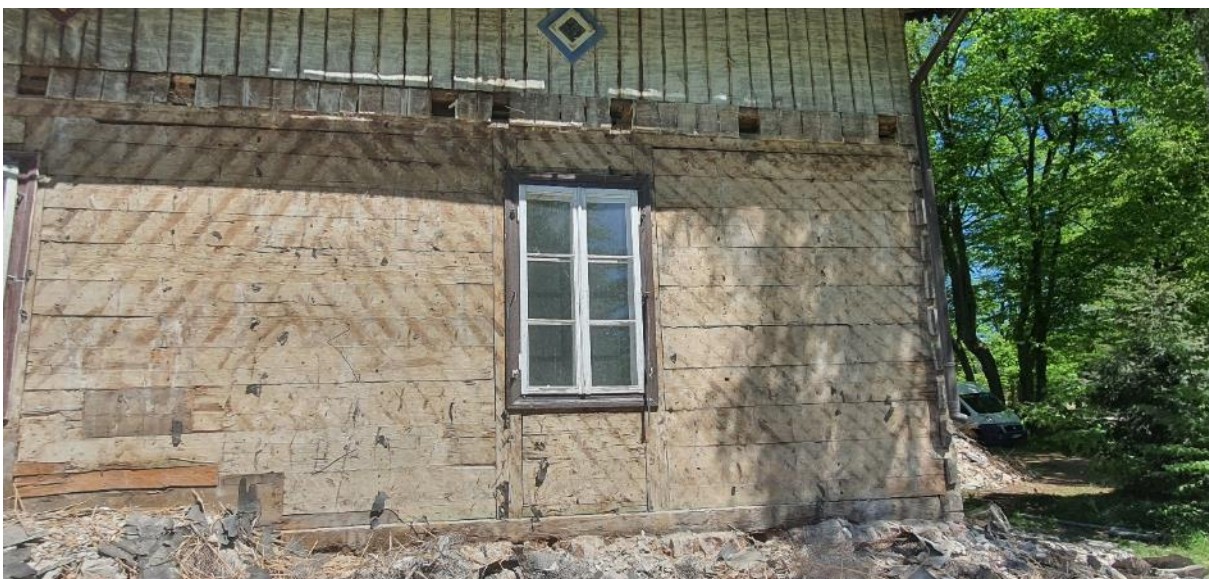

**Figure 4.** Facade walls on the garden side in good condition.

1.4.2. The Lack of Continuity in Wooden Architecture and Construction on the Lands of Modern Poland and the Related Difficulties in Applying Comparative Methods to Identify the Form and Structure of Historic Wooden Buildings

The lack of historical continuity in construction and architecture caused by numerous wars taking place in the territory of modern Poland and the resulting loss of the majority of wooden building structures, often the most valuable, as well as the partitioning of Poland that resulted in different construction policies, different use, and different levels maintenance to existing structures and the new arrangement of borders after World War II,

has resulted in a minimal number of wooden building monuments throughout the country. During diagnostic and conservation studies, when carrying out engineering, architectural, and construction work (strengthening and protection, repair, adaptation, reconstruction), it is usually impossible to relate (according to conservation practice and theory) the applied individual technical solutions to the analogous applied solutions found in existing buildings. The successful implementation of this is rare and is often found in open-air museums (Figures 5–7). A specific answer to this problem appears to be referencing early documentation, which is often published in the form of textbooks, albums, drawings, and lithographs. Interest in wooden architecture in Poland developed notably at the turn of the 19th and 20th centuries. At that time, it was recognized that historic wooden buildings are part of national cultural heritage and thus require special care, documentation, and preservation activities. At the beginning of the twentieth century and in the interwar period, textbook, guidebook, and catalog publications began to be circulated that referred to traditional building techniques, thus presenting historical, current, and modern construction methods, forms, and solutions. The publications and studies made by Professor Jan Sas Zubrzycki and ethnographer and historian Zygmunt Gloger were very significant, and today are an invaluable treasure trove of knowledge about historical wooden buildings that no longer exist and the unique architectural and construction solutions used in them [41–43]. This knowledge is valuable when making decisions related to the preservation, restoration, rehabilitation, or reconstruction of historic wooden buildings while ensuring historical continuity, integrity, and authenticity. The unique architectural and structural solutions used in historic wooden buildings require an individual approach to conservation, engineering, architectural, and construction activities. They are often adapted to local traditions, carpentry art, aesthetic trends, and ornamental traditions and are an expression of the vision of the building's creator, their skills, and their technological and material capabilities (Figure 8).

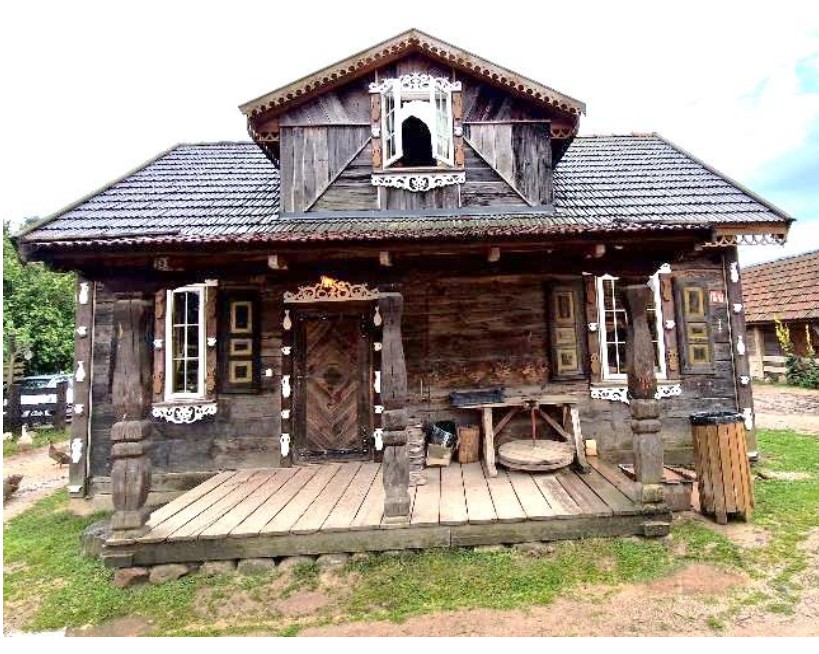
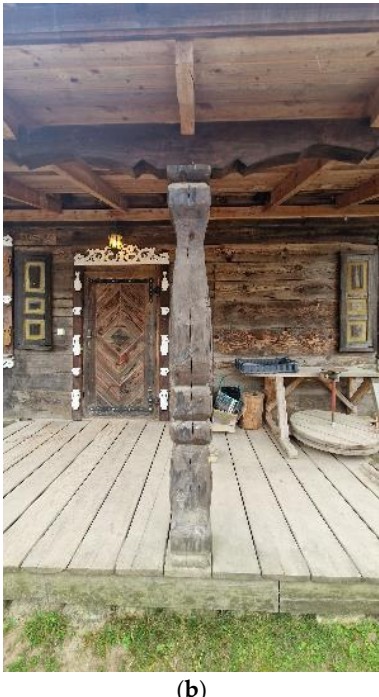

(**a**) (**b**)

**Figure 5.** Examples of a wooden building in an open-air museum: (**a**) façade; (**b**) carved structural elements of the overhang.

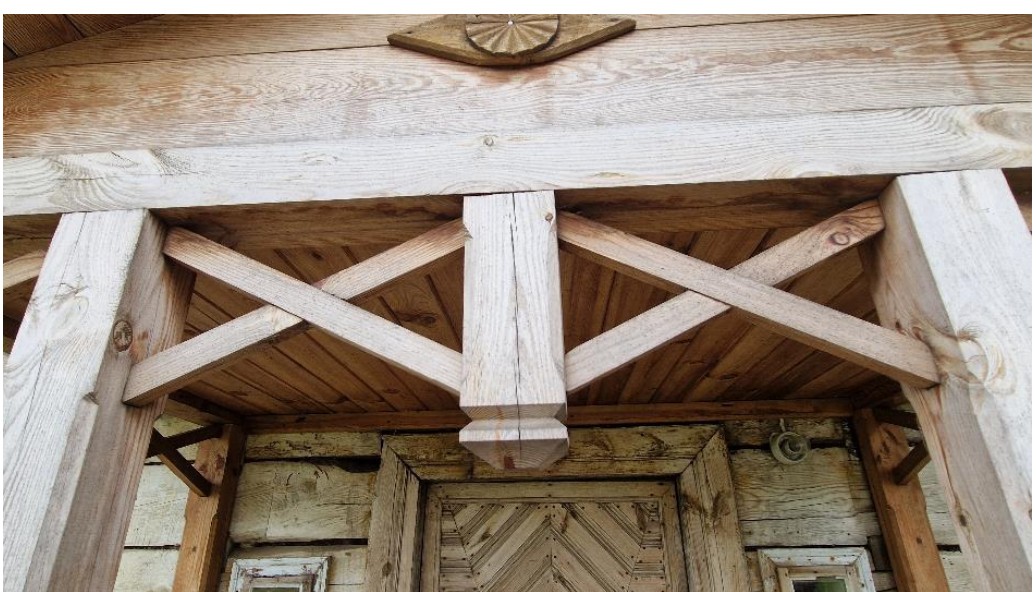

**Figure 6.** Decorative elements of an overhang from a historical wooden house in an open-air museum.

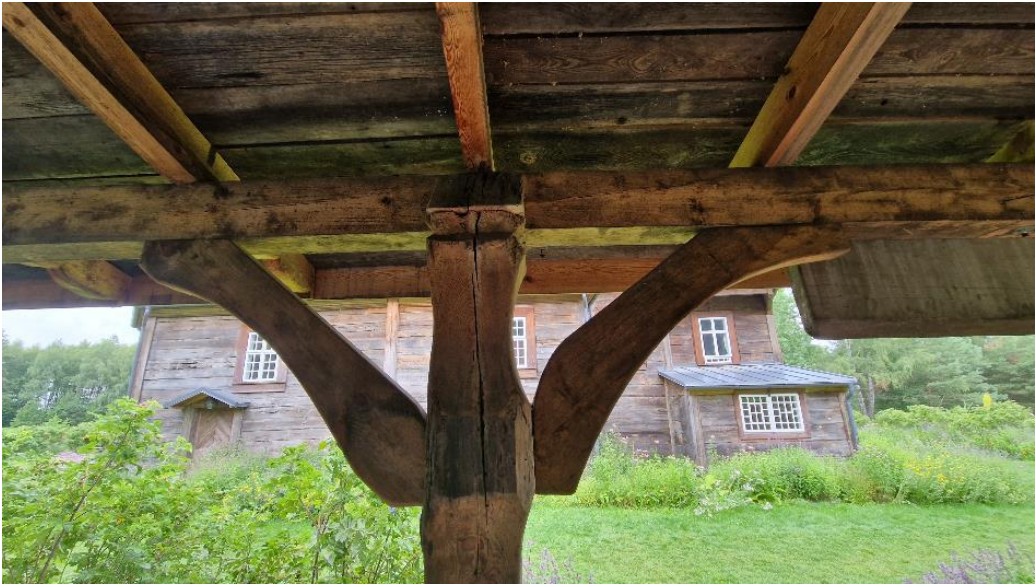

**Figure 7.** Carved structural elements of an overhang from historical wooden house in an open-air museum.

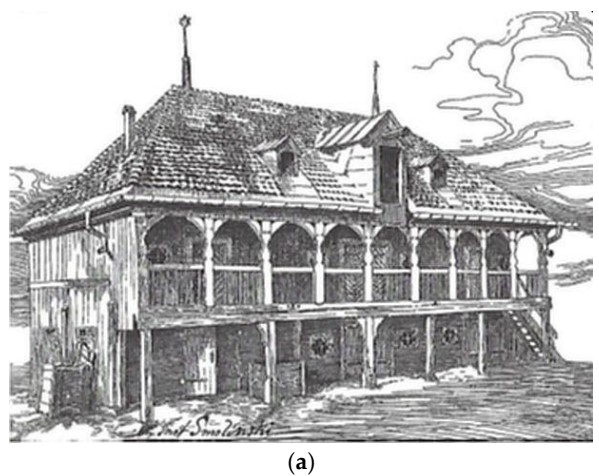 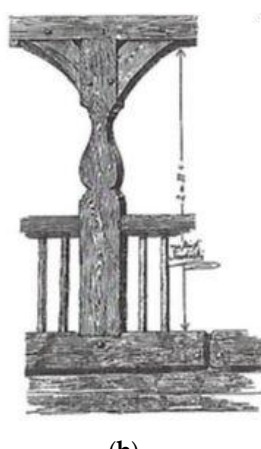

(**a**)                  (**b**)

**Figure 8.** Riverside storehouse on Boleść Street in the Old Town in Warsaw. (**a**) General view; (**b**) details of the upper arcade column of the storehouse [44].

### 1.4.3. Occurrence of Architectural and Structural Solutions That Are Not Used Today

In historic wooden buildings, certain detailed technical, material, or architectural solutions are often unique. Here, we should mention the original and distinctive contemporary construction of ceilings, wooden trusses, stairs, columns, braces, wooden lintels, and the gables of buildings, as well as decorative carving details and carpentry elements for connections and locks found in the main load-bearing components. Unique contemporary solutions for the construction of ceilings have been described and documented in detail by Stanislaw Mielnicki, among others (Figures 9–14) [45]. Some of these ceilings have an original design, which today is unknown or very rarely used (e.g., the "slide-in" ceiling (Figure 10) and ceilings with beams (Figure 14)). Some ceilings used structural elements that are not produced or used in ceilings today (e.g., Bacula strips (Figure 9), reinforced concrete elements (Figure 11), gypsum layers (Figure 12), Tekton slabs (Figure 13), and Suprema boards (Figure 14)). Prof. Jan Tajchman [44] also provided systematics for wooden ceilings. Historical drawings and photographic documentation are essential for comparative studies of degraded, destroyed buildings when seeking to restore important elements and parts of buildings.

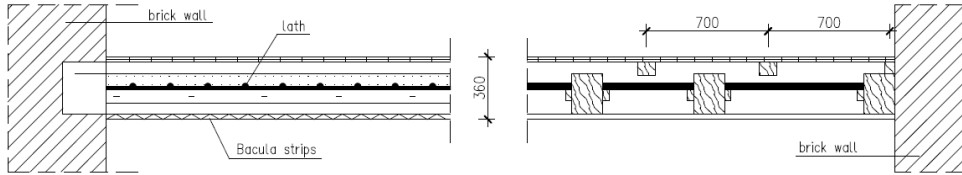

**Figure 9.** Joist–lath (Lviv) ceiling (on the basis of [45]) (p. 101).

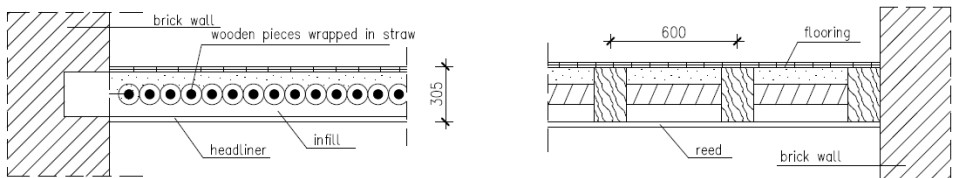

**Figure 10.** "Slide-in" ceiling (on the basis of [45]) (p. 101).

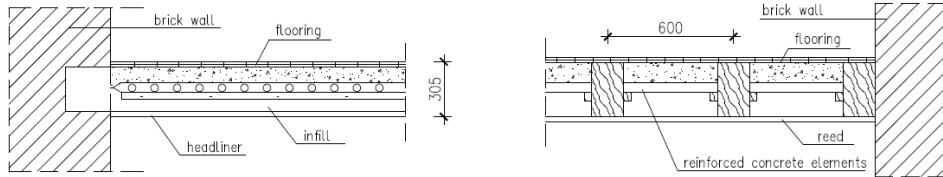

**Figure 11.** Ceiling with massive covering [45] (p. 103).

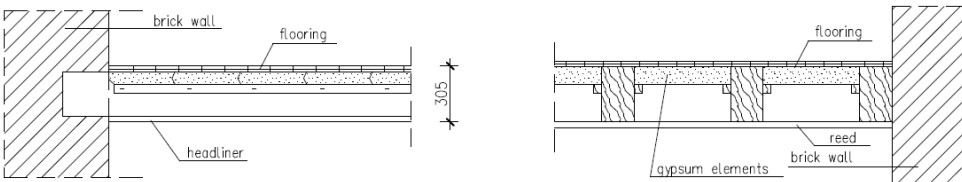

**Figure 12.** Lath ceiling with covering from gypsum layers [45] (p. 103).

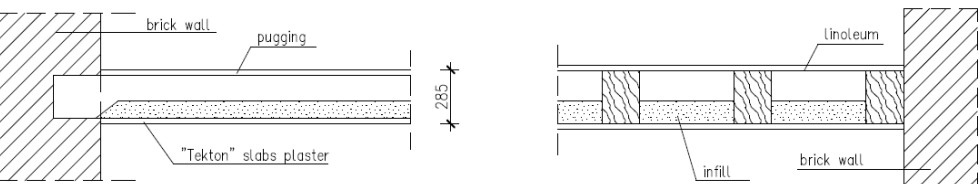

**Figure 13.** Ceiling covered by boards [45] (p. 103).

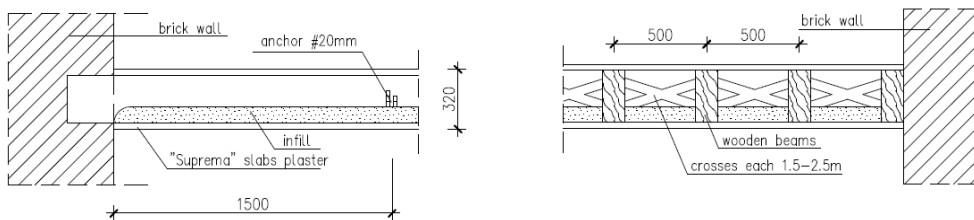

**Figure 14.** Ceilings with wooden beams [45] (p. 103).

Also of interest are methods of installing thermal insulation that are no longer used, including the use of chopped straw properly attached to wooden exterior walls, which was previously common. An example of this solution is shown in Figure 15. Slats arranged in a herringbone pattern attached on the inside of the elevation bearing wall (catfish-patch construction) constitute a frame into which chaff was poured, with the interior surface then covered with boarding.

Another example testifying to the variety of structural solutions used is the structures of roof trusses, which are individualized and adapted to the form and structure of the building. Each truss, especially those covering buildings of elaborate form, such as public buildings, churches, or town halls, is the separate work of an artist, often with unique features. These structures, despite their classical layouts, usually bear the marks of originality and individualization that stem from the creativity and workmanship of the creator, the surrounding tradition, and the influence of local craftsmanship.

As mentioned above, the original structural and architectural solutions used in wooden historical buildings are not widely known, which often causes technical and theoretical problems during various engineering activities.

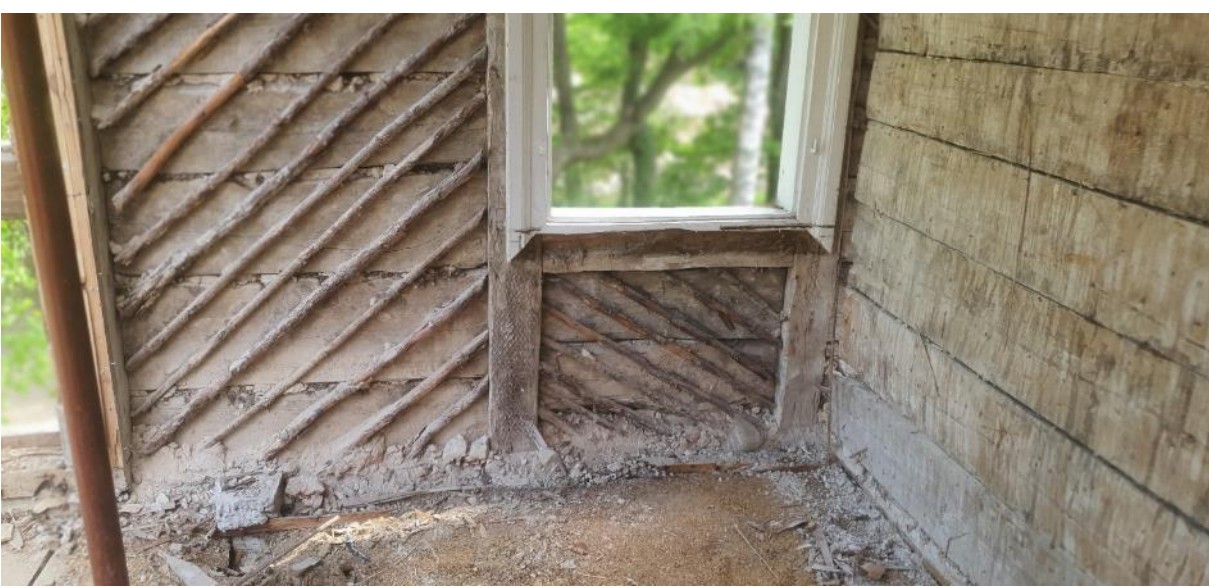

**Figure 15.** Historical thermal insulation installation method.

### 1.4.4. The Need to Replace and Restore Degraded Elements of the Supporting Structure of Wooden Buildings, as Well as Architectural Details and Design Elements

Wooden structures do not degrade and deteriorate uniformly. This is caused by various biological and environmental factors [45,46] and can occur locally or in different parts of the building. In engineering activities on historic wooden buildings, it is often necessary to replace degraded building elements that are part of the structure's load-bearing system. Such elements most often include the foundation, sections of load-bearing walls, ceiling beams, and roof trusses. Replacement of some components is very problematic. This applies primarily to the foundation and the wall-forming elements in the lower part of a building. Replacement of the foundation, if possible, often requires raising the entire building, which can be a performance problem. It should be borne in mind that in this case, the building being lifted should be appropriately strengthened and, if possible, relieved of loads. The same applies to the replacement of wall elements. In extreme cases, it is necessary to demolish a foundation or even the walls in the bottom part of the building to reassemble it with the replaced elements. In this case, it is necessary to remember to carry out appropriate conservation and assembly documentation (a project for the demolition and reassembly of the wooden structure should be drawn up, the individual elements of the structure should be cataloged and numbered, and photographic and graphical documentation should be made). All engineering activities should be fully documented. In case of such actions, it may be crucial to use some form of schedule optimization so as to make it possible to maximize the utility value of the subject of the project, taking into account economic parameters [47,48]. Additionally, the compatibility of materials used and execution techniques should also be considered. The wood from which the replacement elements will be made should be of the same grade as the original and have the same or similar physical and mechanical properties, and the wooden parts should have a moisture content similar to that of the original elements. Architectural elements, artistic details, and finishing elements should be approached analogously.

### 1.4.5. The Need to Carry out Safety Work and Reinforcement Measures in Case the Safety Conditions of the Structure and the Scope of Their Use Is Threatened

Very often, in wooden historical buildings, the safety of the structure and the scope of their use is exceeded due to their poor condition. It should be borne in mind that before starting any activity in the building, including before starting any research and diagnostic activities, the existing structure of the building should be secured and, if necessary, strengthened.

Consideration should be given to the aforementioned problems, as well as to technical issues arising from the peculiarities of historic wooden buildings and their direct impact on the results of the diagnostic process and the resulting conservation and engineering activities.

Taking into account the complexity of the process, it is necessary to take a broader view of the whole issue. It should remembered that there are not only technical problems and issues present in the construction of the building, but also a number of non-technical issues that arise from the principles of sustainable development. In addition to the issues typical of traditional building diagnostics, preparation for the process of conservation and rehabilitation of a historic wooden building should also take into account a whole spectrum of issues that may not have been previously analyzed in it so far.

## 2. Methodology

An assessment of the technical condition of a wooden historic building in the village of Gościeńczyce, near Warsaw, Poland, was carried out using a multi-criteria diagnostics method—Evaluation of the Revitalization Capability of Historic Buildings. This is an original method developed by a team from the Faculty of Civil Engineering at Warsaw University of Technology under the direction of the author. Using this method, the technical and non-technical features of the building are diagnosed, and on this basis revitalization measures are determined, including engineering, structural, and conservation measures. Since it is a multiple evaluation method, it can be used repeatedly during the diagnostic and revitalization processes to check and verify the results achieved at different stages of the activities.

The diagnostic procedure for a historic building must consider conservation recommendations, often limiting the process of research and the degree of intervention chosen for the monument's structure. In the case of historic buildings, an essential element of the completed diagnosis of the technical condition of the monument is a conservation program and often a protection program [49,50]

In terms of implementation of the principles of sustainable development [51] (implying the interdisciplinary nature of the revitalization process [52]), it is necessary to use extended diagnostics that are based on these principles, and capable of analyzing several non-technical problems that ultimately affect the achievement of revitalization goals. The realization of this postulate can be achieved by drawing up an assessment of the revitalization capacity of a historic building and identifying various indicators of the revitalization process. This is a form of interdisciplinary, multifaceted, and multistage diagnostics for existing buildings (also under conservation protection) subject to the revitalization process, in which various sub-indicators defining the process are analyzed.

The revitalization capacity of a building (including a historic building) ($Zd_{rew}$) is a set of characteristics, properties, and states that define the building in terms of its construction, form, function, location, environmental, social, and health values; this capacity also determines the possibility and economic viability of the planned revitalization in a form that takes into account all aspects of revitalization, including the principles of sustainable development [53].

The assessment of revitalization capacity includes issues arising from the principles of sustainable development in construction, significantly expanding the scope of diagnostic studies of a historic building. The principles of sustainable development in construction [50], demonstrating the close connection between economic, social, and environmental issues and construction, should be present in any revitalization process. This follows directly from the definition of both terms. Sustainable development, as stated earlier, defines a development process that, while striving to completely satisfy the diverse needs of the current generation, does not constrain in any way the development potential of future generations, giving them at least the same opportunities as today's generations. This refers to the implementation of thoughtful, planned, interdisciplinary activities (economic, technical, and social, including appropriate use of the environment) in a way that does not cause degradation or irreversible changes to the environment, and also refers to any

economic activities that contribute to the inappropriate distribution of wealth and result in widening economic disparities between different environments and societies and uneven economic growth or living standards. The creation and abandonment of structures in areas in crisis without appropriate corrective measures, degradation in technical, material, and environmental terms, negative effects on social behavior, and causing social and economic problems are ideas contrary to those of sustainable development. Revitalization, defined as complex interdisciplinary activities concerning spatial, technical, and urban changes with the purpose of leading an area out of a crisis, is a process resulting directly from the principles of sustainable development. The realization of revitalization objectives is carried out by restoring the revitalized area to its original utility functions or creating new functions and conditions for its further development.

Sustainable diagnostics (based on the principles of sustainable development) concerning historic buildings should be an interdisciplinary activity. When analyzing the value of a historic building, it is necessary to take into account (in addition to its material value resulting from the technical condition of its structure) architectural and construction systems, functionality and adaptability to new technologies and utility requirements that are currently valid, its intangible value (historical, cultural, aesthetic, environmental), and its impact on the local community in economic, social, and cultural terms. This issue goes far beyond traditional technical diagnostics [25]. A diagram of the interdisciplinary diagnostic procedure is shown in Figure 16.

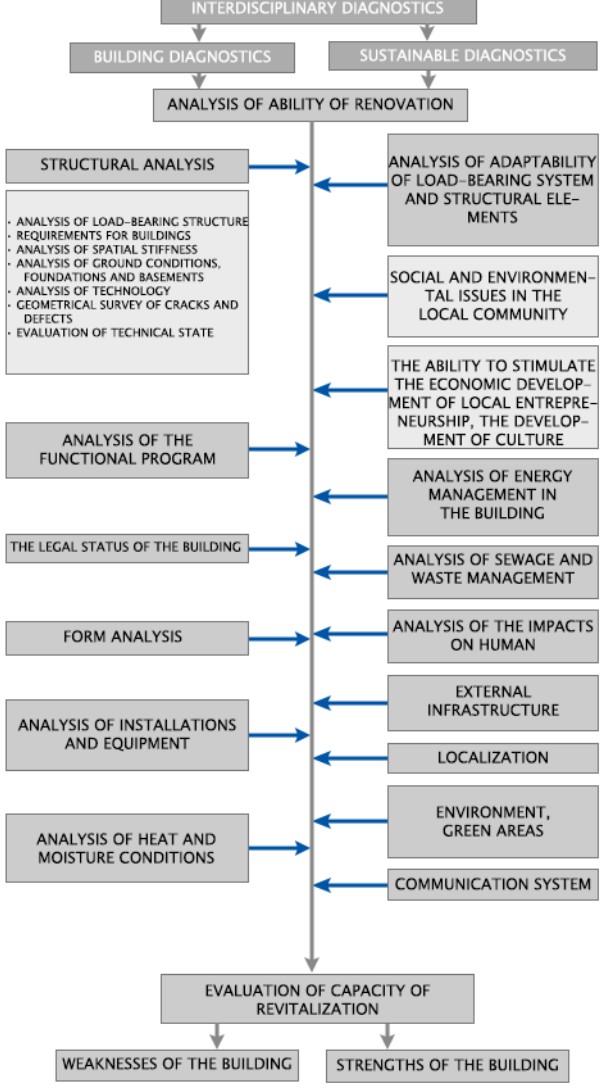

**Figure 16.** A diagram of the interdisciplinary diagnostic procedure.

Using sustainable diagnostics according to the Evaluation of the Revitalization Capacity of Historic Buildings method (which considers interdisciplinary research and analysis), a detailed, accurate evaluation of the revitalization capacity of a historic building can be carried out. To systematize and objectify the way of assessing the revitalization capacity of a historic building, including a wooden building, an assessment algorithm was developed in which, according to specific rules, approx. 100 selected features and conditions are included, which are grouped into 8 groups of interdisciplinary issues. These issues relate to the diagnosis of the building (technical (structural) and non-technical), as well as its historical value, cultural value, environment, and impact on the external environment and local community. These issues, as well as the features and states evaluated in the developed method, are summarized below in Table 1 [20,25,53].

**Table 1.** Issues, features, and states analyzed in the assessment of the revitalization capacity of a historic building.

| **Analysis of Formal and Legal Aspects of the Building** |
|---|
| **Evaluation of Formal and Legal Status** |
| Property matters |
| Land register |
| Mortgage charges |
| Technical documentation |
| Entry in the register of monuments |
| Recognition as a cultural monument |
| Part of the cultural park |
| Protection in the local development plan |
| Entry on the UNESCO World Cultural Heritage list |
| **Analysis of historical and cultural value** |
| **Evaluation of historical and cultural value** |
| Historical value |
| Cultural value |
| **Analysis of building construction aspects** |
| **Architectural evaluation of the building** |
| Analysis of architectural form and system |
| Functional and utility system |
| **Assessment of the technical condition of the building structure** |
| Ground conditions |
| Foundations |
| Walls, pillars |
| Ceilings |
| Roof trusses |
| Roof sheathing |
| Roof covering |
| Balconies |
| **Assessment of the technical condition of the building's facade and finishes** |
| Facade |
| Windows |
| Doors |
| Flashing, gutters, downpipes |
| Cornices, architectural details, decorative elements |
| Plastering and exterior renderings |
| Plastering and interior plastering |
| Historic details |
| Decorative, historic exterior plastering |
| Decorative, antique interior plastering |
| **Evaluation of thermo-humidity conditions** |
| The dampness of external walls |
| The dampness of interior walls |
| Thermal insulation |
| Traces of mold, building fungus, and other biological damage |

**Table 1.** *Cont.*

---

<div align="center">

**Assessment of the condition of installations in the building**
</div>

Electrical installation
Hot water installation
Cold water installation
Sewerage
Ventilation
Air conditioning
Heating
Tiled stoves
Lightning protection system
Fire protection installation

<div align="center">

**Assessment of adaptability**
</div>

Load-bearing structure
Adaptation to new functions
Adaptation to modern requirements
Enhancing user experience
Adaptation to the needs of people with disabilities
The adaptability of structural elements

<div align="center">

**Technological and material assessment**
</div>

Technical condition of used materials, mortars, and construction elements
Availability of materials
Technology

<div align="center">

**Human impact analysis**

**Human impact assessment**
</div>

Harmful effects of mold and mildew
Materials harmful to health
Harmful emissions of materials
Noise
Vibrations

<div align="center">

**Analysis of the external environment**

**Evaluation of the external environment**
</div>

Location
Communication: external transport, opportunities for modernization, adaptation, parking spaces, bicycle and pedestrian routes
Green areas
Integration with green spaces and the environment
Environmental impact
The volume of produced sewage, liquid, and solid waste, the method of treatment and disposal, and the possibility of reduction
Proper, harmless, and pro-environmental lighting
Proper water management
Recycling of materials and waste from the building

<div align="center">

**Energy efficiency analysis**

**Energy efficiency assessment**
</div>

Energy requirements
The energy efficiency of the building partitions
Improving energy efficiency

<div align="center">

**Innovation analysis**

**Innovation assessment**
</div>

Materials
Technologies
Design and technical solutions
Devices and equipment
Solutions in compliance with usage

<div align="center">

**Analysis of the impact on the local community**

**Assessing the impact on the local community**
</div>

Impact on social and societal problems
Economic stimulation and business development
Influence on the development of culture and the arts

---

The values from individual assessments have the character of expert assessments. In order to objectivize them, the author developed guidelines based on current standards, recommendations, and building regulations. In determining the revitalization capacity value, the above features and states, which are the individual aspects of the assessment ($O_i$ represents assessment of the i-th aspect), are evaluated by assigning them the corresponding point rating (from 0 to 5). This determines the values of the sub-assessments assigned to each issue (grouped into 8 groups), which are given different weights because each issue has a different impact on the final assessment.

The Historic Building Revitalization Capability Assessment method contributes to acquiring knowledge about the building being diagnosed, making it possible to conduct a SWOT analysis. To analyze all aspects of revitalization in detail and accurately (using the capabilities of the SWOT analysis), the indicators for partial evaluations of individual groups of issues ($W_i$) are helpful (1). They provide an objective assessment of each group of issues without considering weights. They also provide a value for the total evaluation of a particular group of subjects (by summing up the evaluations of individual aspects, i.e., features and states) in the form of a percentage indicator, taking the maximum total evaluation as 100% (formed from the maximum evaluations of individual aspects in each group of issues).

$$W_i = \frac{\sum O_{crin}}{\sum O_{cmaxn}} \, 100\% \tag{1}$$

where:

$W_i$ represents the index of the i-th group of evaluation issues (i = from 1 to 8);

$O_{crin}$ represents the sub-assessment of the actual n-th aspect in the i-th group of issues;

$O_{cmaxn}$ represents partial evaluation of the maximum of the n-th aspect in the i-th group of issues.

Due to the 8 groups of issues analyzed in the evaluation of the revitalization capacity of a historic building, the following sub-indicators can be distinguished:

- W1—formal and legal evaluation indicator;
- W2—historical and cultural value indicator;
- W3—construction value indicator;
- W4—human impact evaluation indicator;
- W5—external environment evaluation indicator;
- W6—energy efficiency indicator;
- W7—innovation indicator;
- W8—impact on the local community indicator.

The revitalization capacity value of a historic building can be dependent on the value of individual sub-assessment indicators, and can be obtained using formula (2):

$$Zd_{rew} = \sum_i (W_i \cdot g_i) \tag{2}$$

where:

$W_i$ represents an indicator of the i-th group of evaluation issues (i = from 1 to 8);

$g_i$ represents the weight of the indicator used in evaluation of the i-th group of evaluation issues (i = from 1 to 8).

In the above-described method of interdisciplinary diagnostics, an additional path for the analysis of issues is added related to the preservation of monuments and issues relevant to cultural and historical value. The following issues are analyzed:

- Authenticity of the building—integrity, state of preservation, preservation of the original style, stylistic features;
- Artistic value—uniqueness, originality (architectural, plastic, spatial solutions), creator, relationship with the environment, novelty, local tradition, urban context;
- Historical value—association with a historical event or figure, emotional value, local significance, supra-local significance, intangible value;

- Scientific value;
- Interiors—detail in interiors (stucco, parquets, floors, balustrades, cladding, paneling, polychromes, mosaics).

The values of the various indicators are an essential source of knowledge, describing multiple detailed aspects of the process of revitalization of a historic building. This is particularly illustrative in the graphical version and makes it possible to compare the revitalization process of several buildings or several aspects of the revitalization of one building. A building condition index is also calculated, analyzing only structural aspects. The revitalization capacity of the historic building (Zdrew) is the sum of the rehabilitation capacity of the building (Zdreh—based on classical diagnostics commonly used in construction practice) and the sustainability value for the revitalization process (Wzr).

$$Zdrew = Zdreh + Wzr \tag{3}$$

The structural aspects factor W3 and the energy efficiency factor W6 are useful for analyzing the rehabilitation capacity of a building (Zdreh) and relate mainly to the existing technical condition of the building. However, it should be noted that the W6 factor can also show the state that can be achieved after applying revitalization actions, including, for example, thermo-modernization of the building. The other factors (W1, W2, W4, W5, W7, and W8) are useful for analysis of the sustainability value Wzr. When assessing these factors, we can consider the issues to which they relate in two ways. Evaluation of these aspects may refer to both the existing situation of the building and its potential as a result of revitalization actions that can be carried out. A low rating of factors may indicate the bad condition of an existing building, but at the same time may also indicate the high potential that can be realized after applying appropriate revitalization actions.

After assessment of the revitalization capacity of a historic building, a final point score (percentage points) corresponding to the specified ratings is obtained. Point values for the final assessment of Zdrew (total revitalization capacity) and the corresponding descriptive ratings are summarized in Table 2.

**Table 2.** Scale for the final evaluation of revitalization capacity.

| Evaluation of the Revitalization Capacity of a Historic Building | | | |
|---|---|---|---|
| **S.n.** | **Number of Points** | **Evaluation** | **Evaluation Description** |
| 1. | 0–40 | Insufficient | Revitalization not recommended |
| 2. | 40–60 | Sufficient | Complex revitalization, with problems |
| 3. | 60–80 | High | Revitalization recommended |
| 4. | 80–100 | Very high | Revitalization with a high probability of success |

The revitalization capacity of historic buildings determines both the technical condition of the existing building subject to the revitalization process, as well as its interdisciplinary potential that can be exploited in the revitalization process, which thus affects the aggregate value of the building and its role in the process. This role is usually crucial due to the central location of historic buildings in revitalized areas, their functions (original or newly created in the revitalization process), and their impact on the external environment (natural environment and local community).

The Evaluation of the Revitalization Capacity of a Historic Building method is an iterative method, which means that the evaluation can be repeated at different points in the process of revitalizing a building. At that time, slightly different values for individual sub-assessments may be determined, which affects the values of assessment indicators. This is due to the different assumptions that can be made at a given stage of the revitalization process. There are 4 main variants for determining values in the assessment of the revitalization capacity of a historic building and the associated sub-indicators depending on the initial data adopted for determining the assessment:

- Diagnostic (initial) value—determining the state at the time of performing diagnostic activities (means the existing state of the building).
- Potential value—all issues are evaluated in the maximum possible way in a given case (shows the revitalization potential of the building).
- Design value—determines the value of the assessment after the realization of all the assumptions of the building revitalization project (including feasible conservation and engineering measures).
- Realized value—determines the state of the building after the revitalization process has been implemented.

By determining diagnostic value at different stages of the revitalization process, we can verify the correctness and effectiveness of the activities carried out.

The Evaluation of the Revitalization Capacity of a Historic Building method presented above, as a form of interdisciplinary diagnostics, provides the opportunity to obtain the entire spectrum of knowledge about the building, making it possible to fully develop an appropriate conservation and engineering strategy on this basis. Detailed research and analysis of all diagnostic aspects goes well beyond technical issues. This corresponds closely with approaches based on the principles of sustainable development and the ICOMOS guidelines regarding the conservation of historic wooden buildings. It is a method in which data on a historic building—technical, material, conservational, and non-technical—are collected in a structured manner to describe the value of the building and its strengths and weaknesses, threats, and potential opportunities.

It should be noted that interdisciplinary diagnostics are more complicated to perform than standard diagnostics, which identify only technical issues. Interdisciplinary diagnostics require multidisciplinary interaction between experts from different fields. Issues that intersect each other are analyzed. From an engineering point of view, although technical issues that determine the technical condition of the building under examination, its load-bearing structure, and the individual structural elements that make it up are most important, when incorporating approaches based on the principle of sustainable development (including also the principles of conservation) analysis of non-technical issues is necessary to obtain complete knowledge of the building under examination. This is still quite a problem in engineering practice. The understanding of this fact is not widespread, and due to the difficulties in implementing interdisciplinary diagnostics, this approach is still rarely used.

### 3. Results

An example applications of the Evaluation of Revitalization Capacity of Historic Buildings method in interdisciplinary diagnostics, in accordance with the principles of sustainable development, was undertaken on a historic suburban manor house located in the village of Gościeńczyce near Warsaw (Figures 2–4 and 15). The manor was built after the mid-19th century, most likely at the behest of the Tęczynski family. In the 18th century, the local estate was owned by the Pugets—a family of French origin that settled in Poland during the reign of King Augustus II the Strong. The mansion is a free-standing building. The overground part consists of wooden post and plank construction and the basement is made of solid ceramic brick on lime mortar. On the foundation walls, the wall plates are mounted where the wooden structure of the wall is fixed. The building is a two-axis object built on a rectangular plan, measuring 20.5 m × 13.5 m. The building is one-story with a basement and a residential attic and is covered by a gable roof with wide eaves. The ceilings are of beam construction and the roof of collar structure is supported by joinery walls.

In assessing the technical condition of the building, exploratory excavations were made, which showed no material destruction of wall elements or ceiling beams. The central part of the roof had collapsed, indicating damage to the roof trusses, which was confirmed by a visual inspection of the structure from the inside. Laser measurements showed excessive deflection of the ceiling beams in the ceiling over the living room located

on the first floor. The facade and interior plaster showed no excessive moisture, cracks, or significant cracks indicative of damage to the structural elements (locally, damage to the plaster was visible, under which the supporting elements were not damaged).

Applying the author's Evaluation of the Revitalization Capacity of a Historic Building method, the following values for individual variants of the evaluation of the revitalization capacity of a building were obtained:

Diagnostic value of 63.8 points, —Sufficient rating, and recommended revitalization (Figure 17).

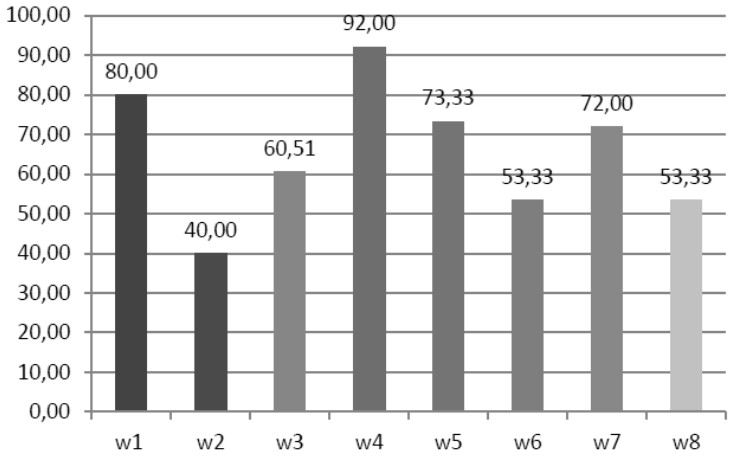

**Figure 17.** Indicator values for partial evaluations: diagnostic value.

W1 = 80.00%, formal and legal evaluation indicator.
W2 = 40.00%, historical and cultural value indicator.
W3 = 60.51%, construction value indicator.
W4 = 92.00%, human impact evaluation indicator.
W5 = 73.33%, external environment evaluation indicator.
W6 = 53.33%, energy efficiency indicator.
W7 = 72.00%, innovation indicator.
W8 = 53.33%, impact on the local community indicator.
Building condition index: 53%.

The technical condition of the entire building was assessed as medium, with the degree of usage of materials ranging from 31–50%.

Potential value of 94.5 points, Sufficient rating, revitalization difficult with problems (Figure 18).

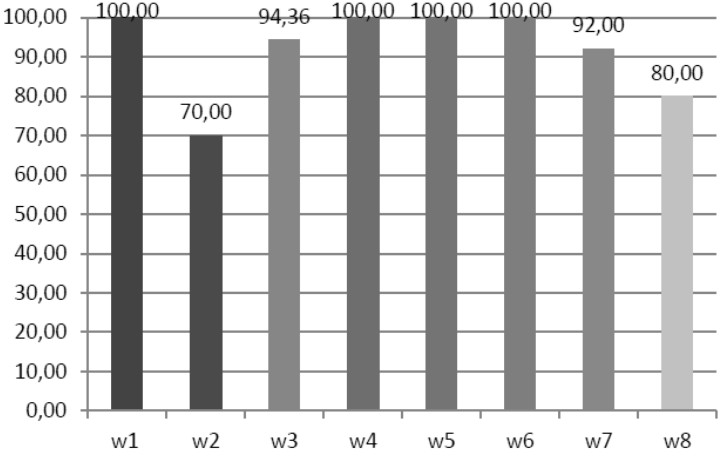

**Figure 18.** Indicator values for partial evaluations: potential value.

W1 = 100.00%, formal and legal evaluation indicator.
W2 = 70.00%, historical and cultural value indicator.
W3 = 94.36%, construction value indicator.
W4 = 100.00%, human impact evaluation indicator.
W5 = 100.00%, external environment evaluation indicator.
W6 = 100.00%, energy efficiency indicator.
W7 = 92.00%, innovation indicator.
W8 = 80.00%, impact on the local community indicator.

Design value of 88.9 points, Sufficient rating, revitalization difficult with problems (Figure 19).

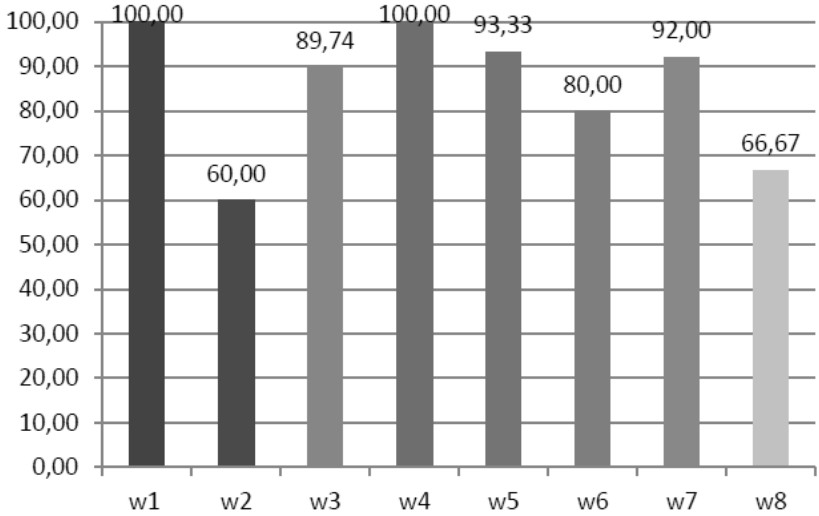

**Figure 19.** Indicator values for partial evaluations: design value.

W1 = 100.00%, formal and legal evaluation indicator.
W2 = 60.00%, historical and cultural value indicator.
W3 = 89.74%, construction value indicator.
W4 = 100.00%, human impact evaluation indicator.
W5 = 93.33%, external environment evaluation indicator.
W6 = 80.00%, energy efficiency indicator.
W7 = 92.00%, innovation indicator.
W8 = 66.67%, impact on the local community indicator.

During the repair and restoration work, it turned out that the actual technical condition of the load-bearing walls was much worse than assessed in the first diagnosis. The wooden beams were completely destroyed in some places, rotten, and fungal. This resulted in a need for a new assessment of the building's revitalization capacity. The new diagnostic value was:

Diagnostic value (second one) of 56.8 points, Sufficient rating, revitalization difficult with problems (Figure 20).

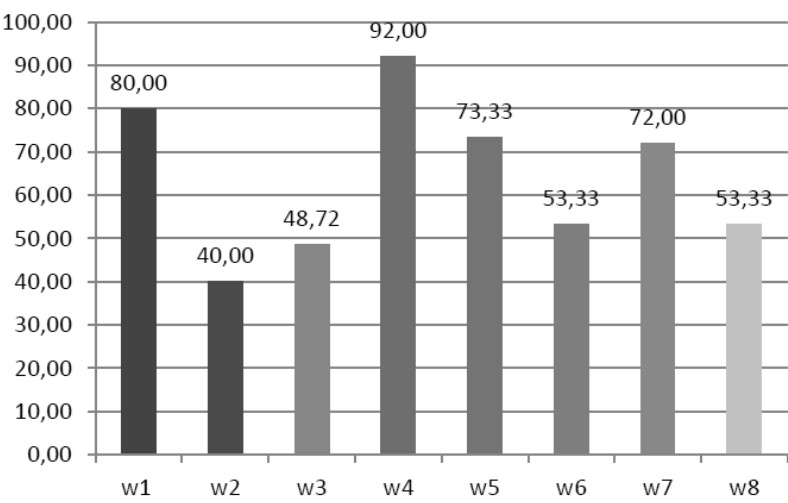

**Figure 20.** Indicator values for partial evaluations: diagnostic value (second one).

W1 = 80.00%, formal and legal evaluation indicator.
W2 = 40.00%, historical and cultural value indicator.
W3 = 48.72%, construction value indicator.
W4 = 92.00%, human impact evaluation indicator.
W5 = 73.33%, external environment evaluation indicator.
W6 = 53.33%, energy efficiency indicator.
W7 = 72.00%, innovation indicator.
W8 = 53.33%, impact on the local community indicator.
Building condition index: 31%.

The technical condition of the entire building was assessed as poor, with the degree of material usage over 50%.

The values of the other options when assessing the revitalization capacity of the building did not change.

## 4. Discussion

Diagnosis of a historic manor house located in Goscinczyce near Warsaw using the original method of this paper confirms its usefulness for interdisciplinary diagnostics of historical buildings, and demonstrates consideration of the principles of sustainable development. In its premise, it analyzes the technical aspects of a building according to traditional technical diagnostics. However, it assumes that in addition to the construction value, it is necessary to take into account the non-technical aspects of historical buildings according to the principles of sustainable development. This includes formal and legal issues (in Poland this is very important due to the remnants left behind and the negligence in this area during communist rule), the historical and cultural value of the building (all recommendations and conservation rules are important here), assessment of the impact on humans, external environment assessment, energy efficiency, the degree of innovation in the adopted design, technological and material solutions, and the impact on the local community. The effects of these analyses can be seen in the form of individual assessments that affect the values of the sub-assessment indicators, which are presented in the charts in the previous section. Looking at the values of the sub-assessment indicators, we get an assessment of these values in the existing state. This provides the basis for initial evaluations that inform the intention to undertake a revitalization investment project based on its feasibility. At the same time, comparing the values of the diagnostic assessment with the values of the potential assessment (Figure 21), the areas of revitalization with the most potential are visible in terms of the individual issues. In these areas, based on local revitalization programs, revitalization measures can be designed that raise the values of the sub-assessment indicators once realized.

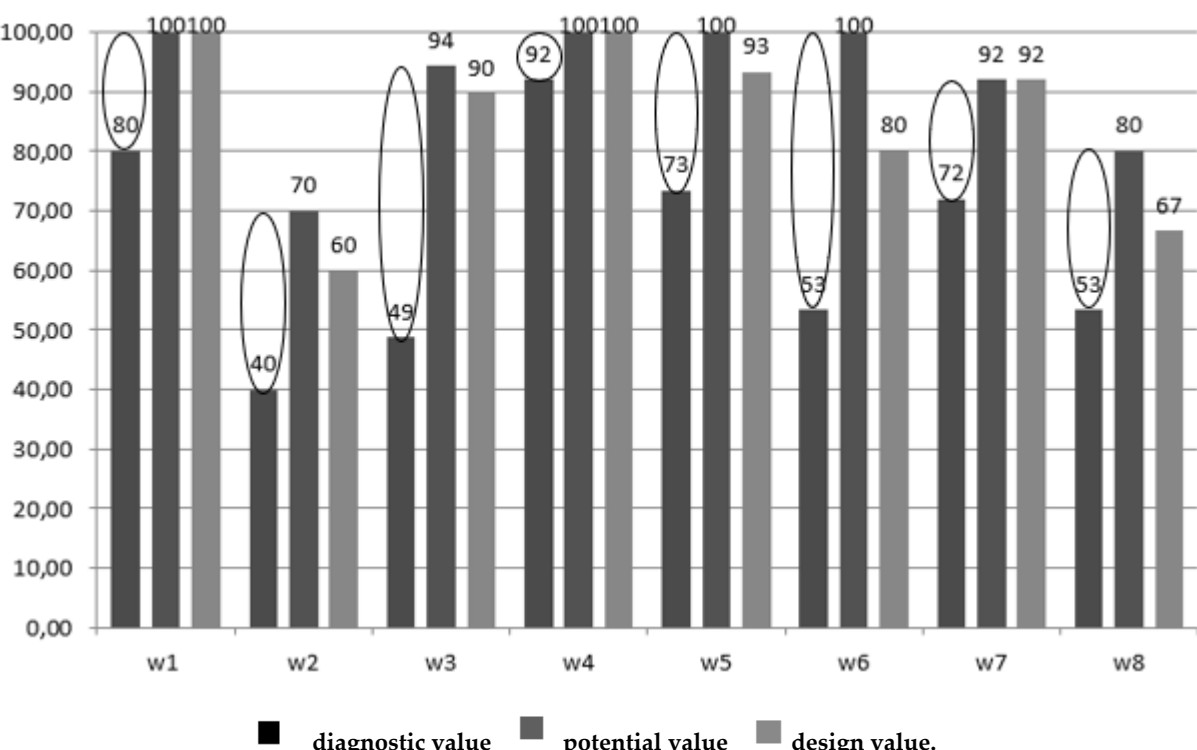

**Figure 21.** Comparison of the values of sub-indicators in terms of potential for revitalization actions and diagnostic values.

The need for a secondary diagnostic evaluation showed that the Evaluation of the Revitalization Capability of a Historic Building method responds appropriately to changes in the values of individual sub-assessments (in this case, the changes were related to building features and were associated with much worse technical conditions in walls and ceilings than those originally diagnosed). The secondary diagnostic assessment properly determined the technical condition of the building and also assessed the revitalization process. In the primary assessment, revitalization was recommended, while in the secondary assessment, it was stated that revitalization was possible with problems, which is in full agreement with the actual state.

It is important to remember that when diagnosing historic wooden buildings, it is necessary to take into account the specifics of these buildings and be aware of all resulting problems and issues discussed in this article. This is important both for the diagnostic process, which prepares conservation and rehabilitation measures, and for the entire process of building revitalization. Further evaluation studies of this method are being carried out so that it can be increasingly adapted to the diagnosis of wooden historic buildings while taking into account their specific characteristics and related issues.

Depending on the level of detail and accuracy in the conducted diagnostic studies, their number, and the interdisciplinary scope, the assessment of revitalization capacity can be carried out at different levels. In a very general approach (this approach can be used in preliminary stages concerning the decision to start the revitalization process), the determination of revitalization capacity can help estimate the value of numerous features that determine both the technical condition of the building, in technical and structural terms, and features resulting from an interdisciplinary approach, based on the principles of sustainable development. Assessment of revitalization capacity carried out in this way can provide an estimate similar to that of SWOT analysis. A properly conducted SWOT analysis shows the following:

- Strengths of the revitalized building, what constitutes its assets, advantages over others, and potential opportunities (this can often be, for example, the good technical condition of a building, good location, and historical or cultural value);
- Weaknesses of the revitalized building, what constitutes its weaknesses, disadvantages, and existing difficulties and problems that need to be overcome in the revitalization process, such as the poor technical condition of a building or structural system and a lack of adequate installations or structural elements forming the load-bearing structure of the building (e.g., cracked and damaged ceilings, damaged foundations);
- Opportunities, including factors that will allow for properly carried out revitalization of the building and everything that creates beneficial changes (e.g., the possibility of changing the function, the possibility of adapting the building to new utility functions, beneficial aspects from the point of view of the local community, economic benefits, social benefits);
- Risks associated with the process of revitalization of the building, its rehabilitation, renovation, modernization, or new uses, and everything that creates the danger of unfavorable changes (e.g., risks associated with changes in utility functions or difficulties associated with the projected rehabilitation of the building).

It can be assumed that strengths and weaknesses are features of the current state, and opportunities and threats are expected future phenomena that may occur during the revitalization process or after its completion during the life of the building.

Evaluations of the various issues analyzed in the Evaluation of the Revitalization Capacity of a Historic Building method are assigned weights, and partial evaluations affect the final evaluation in the diagnostic process.

In order to maximize the objectivity of the criteria for the scoring awarded, auxiliary tables for assessing revitalization capacity were developed. Depending on the group of issues, the numerical results obtained are given specific weights that are assigned to each group. These weights were determined based on an analysis of the systems used to certify buildings in terms of meeting sustainability principles (LEED and BREEAM) as well as the author's own expert experience. Comparing the percentage scores of individual criteria in the total scoring of the two systems of multifaceted LEED and BREEAM certification is difficult since the criteria for evaluating individual aspects are formulated differently. The scopes of issues analyzed in the evaluations of individual criteria, however, partially overlap, which to a certain extent makes it possible to assume certain ranges of values for individual aspects from the total points in the evaluation.

The Evaluation of the Revitalization Capability of a Historic Building method also adopts evaluation criteria not used in other certification systems. They concern issues that are relevant from the point of view of the revitalization process. Despite this, by comparing the issues addressed in the individual criteria for assessing the revitalization capacity of historic buildings with those analyzed in the LEED and BREEAM multi-criteria certification systems, and on the basis of the author's experience and research, it was possible to estimate the ranges of percentage weights relating to the individual assessment criteria. The correct determination of percentage weights, pertaining to individual criteria evaluated using sub-indicators in the revitalization capacity assessment method, is very important for the final assessment. This is because the sub-indicators do not have the same impact on the proper course and end result of the revitalization process. Since the Evaluation of the Revitalization Capacity of a Historic Building method (Zdrew) depends on the evaluation of individual aspects belonging to the eight groups of interdisciplinary issues analyzed in the diagnostic process, it is necessary to determine what impact each group of issues has on the final evaluation. This impact is not equal and depends on the assumptions and goals adopted in the revitalization strategy of the area. The percentage share of each group of issues in the final assessment of revitalization capacity should be adapted to specific revitalization processes and derived from local revitalization programs. Assuming a specific percentage distribution relating to the rehabilitation capacity value of a historic building (Zdreh) and the sustainability value of its revitalization process

(Wzr), whether the planned revitalization activities will be more oriented toward technical activities, rehabilitation and repair activities (very high percentage for the rehabilitation capacity value of the building in terms of the entire assessment of revitalization capacity), or non-technical activities (high percentage for the sustainability value in terms of the entire assessment of revitalization capacity) can be determined.

Based on the assumption that the technical condition of a historic building (the structural aspect of the building which concerns the technical condition of the building as well as its structure, form, architectural–functional layout, and adaptability to modern technical and functional requirements) is the main factor determining the effectiveness of the revitalization of a historic building, it was assumed that the weight of the issues evaluated by this indicator in the overall final scoring of revitalization capacity was 60%. This can be compared to the percentage value of the raw state of the underground part (raw state "0") and the above-ground part of a newly erected building in terms of the cost of all investments of this type. Other aspects of revitalization, affecting primarily the sustainability value of the process (Wzr), have a weight of 40%. As mentioned above, the magnitudes of the impact of the individual issues on the final assessment of revitalization capacity should be adapted to the initial assumptions of the revitalization, which should be established before the start of the revitalization; however, it seems clear that the point value for assessment of the technical condition of the building and other purely technical construction issues should always have a decisive impact on the final assessment value regarding the revitalization capacity of the historic building, and therefore should be more than 50%.

The conducted study shows that application of the Evaluation of the Revitalization Capacity of a Historic Building method to building diagnostics and revitalization processes can be of great importance to investors and contractors involved in these processes. Use of this method can contribute to proper investment decisions that are based on structured, reliable, and interdisciplinary knowledge of the revitalized building. This knowledge, depending on the degree of detail and accuracy in the analysis and research carried out, can be the basis for making specific investment decisions both during the period involving diagnostic processes and the period of actual revitalization activities. A preliminary estimated assessment of the revitalization capacity of a historic building can be a very useful tool for investors when assessing the feasibility of a revitalization process.

The conducted study is also of theoretical importance, as it confirms the possibility of conducting objectified interdisciplinary diagnostics according to systematized research criteria. With this method, it is possible to compare specific technical and non-technical aspects that determine the condition of several buildings located in the revitalized area. Based on a comparison of results from the study of different buildings, and by making certain revitalization assumptions, it is possible to develop a comprehensive program for the revitalization of buildings in the area. This study shows that diagnostics according to the Evaluation of the Revitalization Capacity of a Historic Building method are very suitable for diagnosing wooden historic buildings. Future research developing this method should further tailor individual aspects of the assessment to the specifics of historic wooden buildings. This applies to both technical and non-technical solutions, especially historical and cultural value.

One of the limitations of this method is the impossibility of unambiguously assessing the value of the individual weights of the aspects of the issues analyzed in the diagnostic process. This is due to the possible differing priorities in revitalization processes. This is usually due to assumptions made about the revitalization process by the local community or area owners.

Assessment of the revitalization capacity of a building provides thorough knowledge not only of the existing technical and material value of the building (this range of knowledge can also come from traditional diagnostics of the technical condition of the building, such as expert reports and technical opinions), but also of the intangible value of a building, which is an important added value that is usually overlooked in the diagnostic work

of an engineer. This is often very important from the point of view of investment and business activities. It should be noted that in the case of historic buildings, this value is particularly important. This can increase the value of the building many times over, as the material value of historic buildings is often not substantial due to the advanced process of degradation common in the structure of historic buildings.

## 5. Conclusions

Conservation work, including the entire spectrum of technical construction and architectural activities concerning historic wooden buildings, is a complex, often interdisciplinary activity. In these activities, it is necessary to apply the applicable conservation principles and the principles of sustainable development generally recommended for application. One of the essential elements of the preservation and rehabilitation of wooden structures of historic buildings is appropriate diagnostics preceding these activities. Traditional diagnostics referring only to technical issues do not provide complete knowledge of historic buildings as they do not consider non-technical issues to be important from the point of view of conservation. Additionally, in the context of the revitalization process of a historic building and the area in which it is located, a broader diagnostic is needed that is interdisciplinary and based on the principles of sustainable development. An example of such diagnostics is the method outlined in this paper, which analyzes both the technical aspects of the building under study and the non-technical aspects relevant to the preservation process that can be part of a broader revitalization process. This diagnosis is more demanding than classical technical diagnostics as it requires interdisciplinary cooperation between experts from many trades, which can create implementation problems. Other problems and issues that need to be solved in the process of conservation and rehabilitation of the wooden structure of historic buildings include difficulties in realizing a reliable technical assessment in the diagnostic process of the building; the lack of continuity in wooden architecture and construction on the lands of modern Poland and the associated difficulties in applying comparative methods to identify the form and structure of historic wooden buildings; the presence of architectural and structural solutions that are not used today; the need to replace and restore degraded elements of the load-bearing structure of wooden buildings, architectural details, and elements of decoration; and the need to carry out protective work and strengthening measures in case of threats to the safety of the structure and its use.

In executive and design engineering practice and among many investors, the need to meet requirements related to the principles of the preservation of historic buildings, especially those of wooden construction, is perceived as an additional problem hampering the construction process. There is a lack of widespread awareness regarding the cultural, historical, and aesthetic value derived from the authenticity of a historic building, such as the value of authentic materials, construction techniques, and technical solutions. An interdisciplinary sustainable approach to these issues is rarely applied. This applies to both the diagnostic process and the engineering measures used. This calls for further educational activities in this area that demonstrate the importance of historic buildings, which contribute to national cultural heritage.

**Funding:** This research received no external funding.

**Conflicts of Interest:** The author declares no conflict of interest.

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
