# Peer review of "Problems and Technical Issues in the Diagnosis, Conservation, and Rehabilitation of Structures of Historical Wooden Buildings with a Focus on Wooden Historic Buildings in Poland"

_sustainability, doi:10.3390/su15010510_

Round 1
Reviewer 1 Report
The manuscript discusses problems and technical issues related to the conservation and rehabilitation of wooden structures historic buildings.
Though the manuscript attempts to make contributions to the existing body of knowledge, there are serious issues that need to be addressed.
some of the issues include the following;
1. There is a lack of clear problem statement in the introdution section. This should be addressed by the authors.
2. The significance of the study is not clearly stated.
3. the discussion presented in this manuscript falls short of what is expected in an academic manuscript. This should be greatly improved to make this manuscript suitable for publication.
4. The paper does not contain a conclusion section. This ought not to be so
5. What are the contributions of this study to the body of knowledge? I recommend that the authors state the theoretical and practical contributions of the present study to the body of knowledge.
6. The limitations of the study and future study directions should be stated in the paper.
7. There should be a methodology section for the paper. Currently the paper has no methodology section.
Author Response
A revised version taking into account the comments has been created.Reviewer 2 Report
The issue of diagnosis, conservation and rehabilitation of historical timber structures is of great interest and is still much discussed in the literature.
The attempt to interpret the principles dictated by the ICOMOS guidelines is interesting, but it would also be necessary to evaluate its applications to significant examples of the built heritage.
The problem is that, in the paper, all aspects related to the above issue is not deeply argued. Moreover, the lack of specific applications and results make the paper too descriptive. No technical issue has been discussed or suggested.
The Historic Building Revitalization Capability Assessment Method would be a good idea but should be further explored.
The paper is strongly referred to Poland houses but timber heritage all over the world is very different and constituted by churches, magnificent roofs and so on. I strongly suggest changing the title in a more appropriate sense.
References are mainly self-cited. There are a lot of examples in literature referred to diagnosis, they would be worth considering.
In summary, the descriptive part of the difficulties and needs is too long, while the conclusions are lacking because there are not enough applications to validate the conclusions.
I felt confident you can improve your paper taking into account the above major considerations.
best regards
Author Response
A revised version taking into account the comments has been created.
Reviewer 3 Report
This work is some interesting for diagnosis, conservation, and rehabilitation of structures of historical wooden buildings. However, there are some questions as follows:
1. In Line 383, “All repairs to wooden structural elements should be carried out traditionally, using carpentry techniques”, It is pointed out here that all rehabilitation processes are to be performed using traditional methods, based on practical experience whether modern methods have been considered better than traditional methods after optimization;
2. The resolution of Figure 2 should be increased;
3. Please standardize the format of references.
Author Response

(The authors gave the same response as above.)

Round 2
Reviewer 1 Report
Thank you for considering some of my comments. However, most of my concerns have not been addressed.
Some of my comments are listed below:
1. Section 1.1 of the paper does not contain a single reference. This is quite surprising.
2. The conclusion section should be after the discussion and not the other way round.
3. the discussion presented in this manuscript falls short of what is expected in an academic manuscript. This should be greatly improved to make this manuscript suitable for publication.
4. What are the contributions of this study to the body of knowledge? I recommend that the authors state the theoretical and practical contributions of the present study to the body of knowledge.
5. The limitations of the study and future study directions should be stated in the paper.
6. There should be a methodology section for the paper. Currently the paper has no methodology section
Author Response
Thanks very much for your comments. I try to impove the work so:
1. some reference added 2. The conclusion section after the discussion section 3. the discussion improved 4. some contributions and limitations added 5. methodology section added
Reviewer 2 Report
Minor revisions
In general, the additional explanations (in red) are very verbose. Whenever possible I would suggest synthesizing.
Line 56-76
present some repetitions as technical and non-technical criteria that could be avoided
Line 263
1. Inspection, survey, and documentation. In the par. it is strongly recommended to include at least the following references, the first on diagnosis of historical timber structures and the second on reconstruction of historical timber structures in HBIM environment:
Santini S., Baggio C., Sguerri L. “Sustainable interventions: Conservation of Old Timber Roof of Michelangelo’s Cloister in Diocletian’s Baths”. International Journal of Architectural Heritage, 2021
Santini S.-, Borghese V., Micheli M., Paz E.O. “Sustainable Recovery of Architectural Heritage: The Experience of a Worksite School in San Salvador”. Sustainability, 2022, 14(2), 608
Line 306-307
diagnostic excavations of structural elements (?)
Line 420
Wood corrosion. In my opinion, corrosion is typically referred to steel. Timber are subjected to other mechanisms of degradation.
Lines 442-446
It is suggested to better clarify the meaning of what is impossible to relate.
Figures 9-14 and 16
The images are not very clear.
Author Response
Thanks very much for comments. I try to improve my work so:
- corrections in language are made ( corrosion, exacavation etc)
- references added 3. clarification in necessary points 4. I will try to finally upload drawings of better resolution
Round 3
Reviewer 1 Report
Thank you for addressing my earlier concerns.